# Single-cell analysis reveals lasting immunological consequences of influenza infection and respiratory immunization in the pig lung

Andrew Muir[1]*, Basudev Paudyal[2], Selma Schmidt[2], Ehsan Sedaghat-Rostami[2], Soumendu Chakravarti[2], Sonia Villanueva-Hernández[2¤], Katy Moffat[2], Noemi Polo[2], Nicos Angelopoulos[2], Anna Schmidt[3,4], Matthias Tenbusch[3,4], Graham Freimanis[2], Wilhelm Gerner [2☯‡]*, Arianne C. Richard[1☯‡]*, Elma Tchilian[2☯‡]*

1 Immunology Programme, The Babraham Institute, Cambridge, United Kingdom, 2 The Pirbright Institute, Pirbright, United Kingdom, 3 Virologisches Institut-Klinische und Molekulare Virologie, Universitätsklinikum Erlangen, Friedrich-Alexander-Universität (FAU) Erlangen-Nürnberg, Erlangen, Germany, 4 FAU Profilzentrum Immunmedizin (FAU I-MED), Friedrich-Alexander-Universität (FAU) Erlangen-Nürnberg, Erlangen, Germany

☯ These authors contributed equally to this work.
¤ Current address: IrsiCaixa, Badalona, Barcelona, Spain
‡ These authors share senior authorship.
* Andrew.Muir@babraham.ac.uk (AM); wilhelm.gerner@pirbright.ac.uk (WG); Arianne.Richard@babraham.ac.uk (ACR); elma.tchilian@pirbright.ac.uk (ET)

**Data Availability Statement:** All R scripts used for processing of the scRNA-seq data are available at: https://github.com/muir-a/PirbrightBAL-muir.

## Abstract

The pig is a natural host for influenza viruses and integrally involved in virus evolution through interspecies transmissions between humans and swine. Swine have many physiological, anatomical, and immunological similarities to humans, and are an excellent model for human influenza. Here, we employed single cell RNA-sequencing (scRNA-seq) and flow cytometry to characterize the major leukocyte subsets in bronchoalveolar lavage (BAL), twenty-one days after H1N1pdm09 infection or respiratory immunization with an adenoviral vector vaccine expressing hemagglutinin and nucleoprotein with or without IL-1β. Mapping scRNA-seq clusters from BAL onto those previously described in peripheral blood facilitated annotation and highlighted differences between tissue resident and circulating immune cells. ScRNA-seq data and functional assays revealed lasting impacts of immune challenge on BAL populations. First, mucosal administration of IL-1β reduced the number of functionally active Treg cells. Second, influenza infection upregulated IFI6 in BAL cells and decreased their susceptibility to virus replication *in vitro*. Our data provide a reference map of porcine BAL cells and reveal lasting immunological consequences of influenza infection and respiratory immunization in a highly relevant large animal model for respiratory virus infection.

## Author summary

Pigs and humans have a similar anatomy and physiology. In humans, cells from lung-washes are used to study immune responses, and it was shown that these cells are crucial

Fastq and h5 files from sequencing are available at: GEO (GSE249866) https://www.ncbi.nlm.nih.gov/geo/query/acc.cgi?acc=GSE249866. All flow cytometry data is provided in the manuscript and supplementary figures.

**Funding:** This work was supported by the Biotechnology and Biological Sciences Research Council (BBSRC) Strategic Program and Core Capability Grants to The Pirbright Institute (BBS/E/I/00007031, BBS/E/I/00007037, BBS/E/PI/230001A, The Pirbright Institute Development Grant [IDG], 2021) and BBSRC grants to the Babraham Institute (BBS/E/B/000C0427, Babraham Institute Development [IDG] Grant 2021, Babraham Institute Core Capability Grant). A VetBioNet award, under the European Union's Horizon 2020 research and innovation programme, grant agreement No 731014 (Ref: VBN_20_49) was granted to MT. The authors would also like to acknowledge the Bioinformatics, Sequencing, Proteomics and Pirbright Flow Cytometry facility and support through the Core capability grant (BBS/E/I/00007039, and BBS/E/PI/23NB0003). AR is supported by an MRC Career Development Award (MR/W016303/1). The Biological Research Facility and Immunological Toolbox (generation of CD161 mAb) are supported by funding from UKRI-BBSRC (BB/CCG2270/1). The funders had no role in study design, data collection and analysis, decision to publish, or preparation of the manuscript.

**Competing interests:** The authors have declared that no competing interests exist.

in protection against respiratory diseases such as influenza and COVID-19. To better understand lung immunity, we compared genes expressed in cells of pig lung-wash to white blood cells, providing an atlas for future studies of immunity in the lung. We also tested a vaccine given to the lung containing IL-1β, a strong immune activator that protects mice against influenza virus infection. However, although IL-1β increased pig immune responses, it did not protect pigs against infection. We also showed that IL-1β reduced the number of immune cells that dampen immune responses (regulatory T cells). In addition, we demonstrated increased expression of a protein, IFN alpha-inducible protein 6 (IFI6), 21 days after infection, showing that while immune cells in the lung have common properties, invading organisms can have a long-lasting impact. Our study elucidates why some vaccines fail despite inducing powerful immune responses, emphasizes the need for caution when applying results from small animals like mice to humans, and indicates the importance of the pig as a model to study disease in humans and livestock.

## Introduction

Respiratory viruses pose significant global health threats, with influenza and coronaviruses responsible for major human epidemics and pandemics. The immune responses to these respiratory pathogens are initiated within the respiratory tract. Alveolar macrophages and dendritic cells continuously survey and sample the respiratory lumen, contributing to the induction of immune responses by secretion of cytokines and chemokines. These inflammatory mediators recruit innate cells including neutrophils and natural killer cells, which possess the ability to kill virus-infected cells. Adaptive immune responses initiated in the local draining lymph nodes are mediated by CD4 and CD8 T cells that enter the lung. These cells recognize relatively conserved internal viral proteins and can mediate cross protection against infections with serologically distinct virus strains [1,2]. A proportion of these effector T cells remains in the lung to become tissue resident memory cells (TRM) [3,4]. TRM can mount a rapid protective immune response on antigen encounter and have been shown to be critical in heterotypic cross protection against influenza in mice [5,6]. In contrast, the best correlate of type-specific adaptive immunity is antibody, and multiple lung-resident B cell (BRM) subsets exhibiting spatial and phenotypic diversity have been described [7–9].

Understanding how an immune response is induced and maintained in the lung is crucial for the design of more effective vaccines that provide long-lasting immunity. Respiratory immune responses are highly localized, and direct investigation of lung cells and fluids, rather than assessment of correlates in blood, is essential [10]. Mucosal immune responses following antigen exposure have been extensively characterized in mouse models of infection, helping to define phenotypic, functional and transcriptional features of lung immune cells [11–14]. However, investigating respiratory immune responses in humans is more challenging, although techniques allowing direct sampling of the respiratory tract are constantly improving. One common approach is the collection of bronchoalveolar lavage (BAL), containing cells and fluid from the airways and alveoli. BAL is enriched with TRM, and it has been shown that the presence of respiratory syncytial virus-specific CD8 TRM in the BAL correlates with reduced symptoms and viral load in human challenge studies [15]. High dimensional phenotypic, transcriptomic, and functional profiling of immune responses in paired BAL and blood samples from COVID-19 patients revealed key roles for airway T cells, monocytes and macrophages associated with disease outcome [16]. Single-cell transcriptome profiling (scRNA-seq) has enabled high resolution mapping of cellular heterogeneity, developmental trajectories, and

activation status of BAL cells, revealing how these responses differ across species and between different infections [17–24]. These studies offer insight for the development of targeted therapies and vaccines.

In recent years it has become clear that targeting TRM is a promising vaccine strategy in inducing long term protection against respiratory viruses. Intranasal administration of a live-attenuated influenza vaccine generated long-term virus-specific TRM in the lungs of mice, which mediated cross protective heterotypic immunity in the absence of neutralizing antibodies, suggesting that for optimal induction of heterotypic immunity, virus infection of the lungs or immunization of the respiratory tract is required [14,25–27]. Recent immunization strategies for achieving heterotypic immunity via TRM include pulmonary delivery of influenza vaccine candidates, targeting conserved internal virus proteins such as the nucleoprotein (NP). These strategies prevent infection, limit viral replication or reduce disease severity in the absence of cross-neutralising antibodies in pre-clinical models [5,28–30].

Pigs are an important livestock species and serve as a significant biomedical model for studying human disease [31]. Pigs have a longer life span, and are genetically, immunologically, physiologically, and anatomically more like humans than small laboratory animals [32,33]. Pigs are also an important, natural, large animal host for influenza A viruses and are infected by the same subtypes of H1N1 and H3N2 viruses as humans [34]. Pigs can also be a source of new viruses potentially capable of initiating pandemics [35,36]. Pigs exhibit similar clinical manifestations and pathogenesis to humans when infected with influenza viruses making them an excellent model to study immunity to influenza [37]. This similarity extends to the lobar and bronchial anatomy, as well as the histological structure of pig lungs which closely resembles that in humans. Additionally, pigs exhibit a comparable distribution of sialic acid receptors in the respiratory tract [38]. We have established a robust influenza challenge pig model to study immunity to influenza and evaluate the efficacy of novel vaccines and therapeutics [39–42]. We have further shown that pig immune responses following infection with H1N1pdm09 influenza virus are similar to those induced in humans [43].

Here we used scRNA-seq to analyse alveolar macrophages, CD4 T cells, CD8 T cells and B cells from the BAL of inbred Babraham pigs to define cellular states of homeostasis and activation following H1N1pdm09 (pH1N1) infection. We also characterized these cells following respiratory immunization with an adenoviral vector expressing hemagglutinin and nucleoprotein in the presence or absence of IL-1β, which is a potent mucosal adjuvant that significantly increases antibody and T cell responses. This high-resolution analysis of BAL cells provides insights into porcine resident leukocytes in the lung environment and the changes they undergo following infection and immunization.

## Results

### Experimental design and cell sorting of BAL

In the present study, we utilized samples from a previous animal experiment described in [44]. The experiment involved four groups of inbred Babraham pigs: one group (n = 5) was infected intranasally with pH1N1 influenza virus, another group (n = 5) received intranasal immunization with a recombinant adenoviral vector expressing hemagglutinin (HA) and nucleoprotein (NP) Ad-HA/NP, and the third group (n = 5) was immunized intranasally with Ad-HA/NP and a recombinant adenoviral vector encoding porcine IL-1β (Ad-HA/NP+Ad-IL-1β). Controls (n = 3) received PBS intranasally (**Fig 1A**). The animals were culled 21 days post immunization or infection and tissues harvested for immunological and virological analysis. In the previous study, immune responses of these groups were extensively analyzed and showed that IL-1β acts as a potent mucosal adjuvant, increasing antibody responses, proportions of plasma

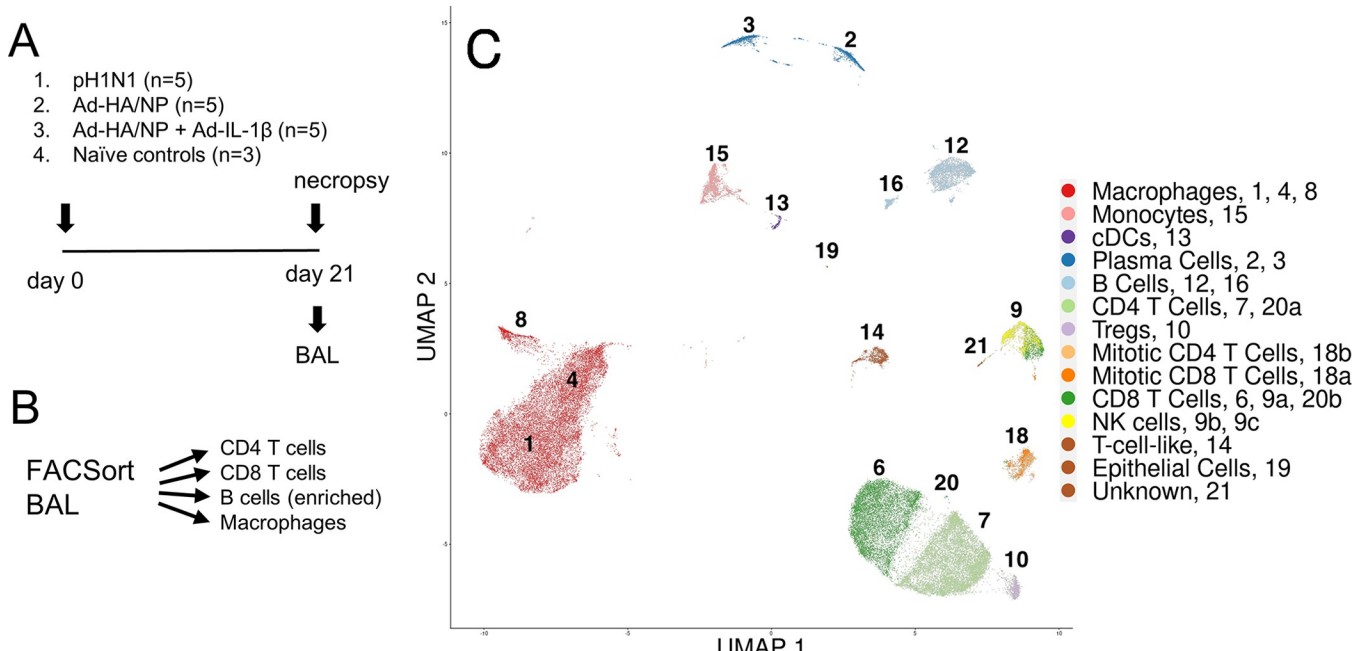

**Fig 1. Experimental design and UMAP clustering of BAL immune cells.** (A) Design of pig immunization/infection experiment. Total number of pigs per treatment is given. For scRNA-seq, 3 pigs were randomly selected for treatments 1, 2 and 3. (B) Graphical overview of applied sorting strategy. (C) Uniform manifold approximation and projection (UMAP) of all cells that passed quality control from pig BAL scRNA-seq analysis, colored by cell-type and labelled with cluster numbers.

and activated B cells, cytokine production of T cells and NP tetramer-specific CD8 T cells in the Ad-HA/NP+Ad-IL-1β group [44].

Given the well-defined nature of these samples we chose to perform scRNA-seq analysis on leukocyte populations sorted from the BAL. We were particularly interested in the BAL as T cells in this niche are inaccessible to intravenous CD3 antibody and have a phenotype characteristic of TRM in other species [45,46]. As the cellular composition of BAL samples is strongly dominated by alveolar macrophages, we applied cell sorting to isolate macrophages, CD4 T, CD8 T and B cells from 3 animals per group and subjected equal proportions of these cells (12,000 cells each) to scRNA-seq (**Figs 1B and S1A**). Cell partitioning and library preparation were performed separately for each sorted and re-pooled BAL sample from 12 pigs (3 animals per group).

## Pig BAL scRNA-seq identifies typical immune cell populations

After combining processed single-cell transcriptomes from all 11 samples that passed quality control checks, graph-based clustering identified 21 distinct cell clusters. Of these, three clusters were excluded for being of low quality or under suspicion of containing doublets. Three clusters were additionally split into subclusters to separate CD4 from CD8 T cells or split T cells from non-T cells, creating a final dataset with 22 cell clusters for annotation. To annotate each cluster with conventional cell-type names, marker genes defining each cluster were extracted and each cluster was screened for expression of a list of typical defining marker genes (**S1 Table, S1B Fig**). From this information, each cluster was manually annotated with an established cell-type name (**Fig 1C**).

Most clusters were found to be comprised of well-defined cell-types, such as macrophages, conventional dendritic cells (cDCs), monocytes, plasma cells, B cells and T cells, though

particular subclusters presented more niche and complex phenotypes. These included 18a and 18b, which were annotated as 'mitotic T cells' based on their strong expression of E2F1 and E2F2, and very small clusters of CD4 and CD8 T cells (20a and 20b, with 18 and 8 cells, respectively) that differed from the major CD4 and CD8 T cell clusters (7 and 6, respectively) by expressing higher levels of genes associated with naïve and/or central memory T cell states (e.g. BACH2, SELL, LEF1) and lower levels of genes encoding cytoskeletal and adhesion molecules (e.g. LGALS1, S100A4, VIM) previously found to characterize lung-localized human T cells [17].

Cluster 14 expressed T cell-related genes in an atypical pattern: strong expression of CD28 and CD5, with low expression of CD4, CD8A and CD8B, bifurcated expression of CD3D and CD3E, and very high expression of EOMES. Pending further investigation, cluster 14 has been annotated as 'T cell like'.

Cluster 21 localises near identified T cell clusters in both UMAP (**Fig 1C**) and TSNE (**S1D Fig**) space. However, cluster 21 does not express any of the conventional αβ T cell markers: CD28, CD5, CD3D, CD3E, CD4, CD8A, or CD8B. Expressed T cell-related genes include GATA3, IL1RL1, IL7R and KIT, suggesting that these cells may be innate lymphoid cells, but without corroborating evidence, we have currently classified these cells as 'unknown'.

Cluster 19 showed strong expression of ACTG1, GPR37, NAPSA and SFTPA1, typically expressed by mucus producing epithelial cells. Cluster 9 and its subclusters were enriched for genes typically associated with NK cells such as KLRK1, KLRB1, KIT and NCR1. As our cell sorting strategy was not intended to capture NK cells, NK cell-like T cells or epithelial cells, clusters 9 and 19 risked presenting a biased representation of these larger cell populations. We therefore excluded clusters 9 and 19, leaving 18 clusters for further analysis.

## Confirmation of scRNA-seq derived cell types by flow cytometry

We aimed to confirm cell types identified by scRNA-seq by flow cytometry of the same cryopreserved samples (**Fig 2**). By using an antibody panel designed for phenotyping of myeloid cells (**S2 Table**) and gating on FSC-A$^{high}$SSC-A$^{high}$ cells (**S2A Fig**), we identified a prominent population of CD172a$^{high}$CD163$^+$CD14$^{-/dim}$CD16$^+$ macrophages (**Fig 2A**), which represent scRNA-seq clusters 1, 4 and 8. Within CD163$^-$ cells, we identified a CD14$^+$CD16$^+$ subset, representing putative monocytes identified in scRNA-seq cluster 15. CD14$^-$ cells were further analysed for CD172a/CADM1 co-expression which allowed the identification of CD172$^{dim}$CADM1$^{high}$ subset and a CD172a$^{high}$CADM1$^+$ subset, representing cDC1 and cDC2 cells, respectively [47], and identified in cluster 13 by scRNA-seq.

The presence of B cells and plasma cells was also confirmed by flow cytometry (**Fig 2B**) using a specifically designed B cell panel (**S2 Table**). CD79α$^+$ cells contained a prominent subset of IRF4$^{high}$Blimp-1$^+$ plasma cells, which were also Pax-5$^-$ and thus resembled clusters 2 and 3 of the scRNA-seq data. CD79α$^+$IRF4$^{dim}$Blimp-1$^-$ cells had variable levels of Pax-5 but overall likely represent B cell clusters 12 and 16 from the scRNA-seq data.

CD4 and CD8 T cells were investigated by a dedicated T cell panel (**S2 Table**). CD3$^+$CD4$^+$ T cells were separated based on Foxp3 expression (**Fig 2C**). Both CD25$^{high}$Foxp3$^+$ putative regulatory T cells (Tregs) and CD25$^{dim}$Foxp3$^-$ non-Treg CD4 T cells were largely CD8α$^{dim}$CCR7$^-$Bcl-6$^-$Eomes$^-$ but expressed low levels of T-bet. With these phenotypes, they resembled scRNA-seq clusters 10 and 7, respectively. We also identified a subset of Ki-67-expressing CD4 T cells, which may represent cluster 18b annotated as "mitotic" CD4 T cells. CD8 T cells were identified by a CD3$^+$CD8αβ$^+$ phenotype (**Fig 2D**). After exclusion of γδ T cells, the majority of remaining CD8 T cells were CD161$^-$NKp46$^-$perforin$^-$, resembling CD8 T cell cluster 6 identified by scRNA-seq. Within this phenotype, some cells expressed Ki-67,

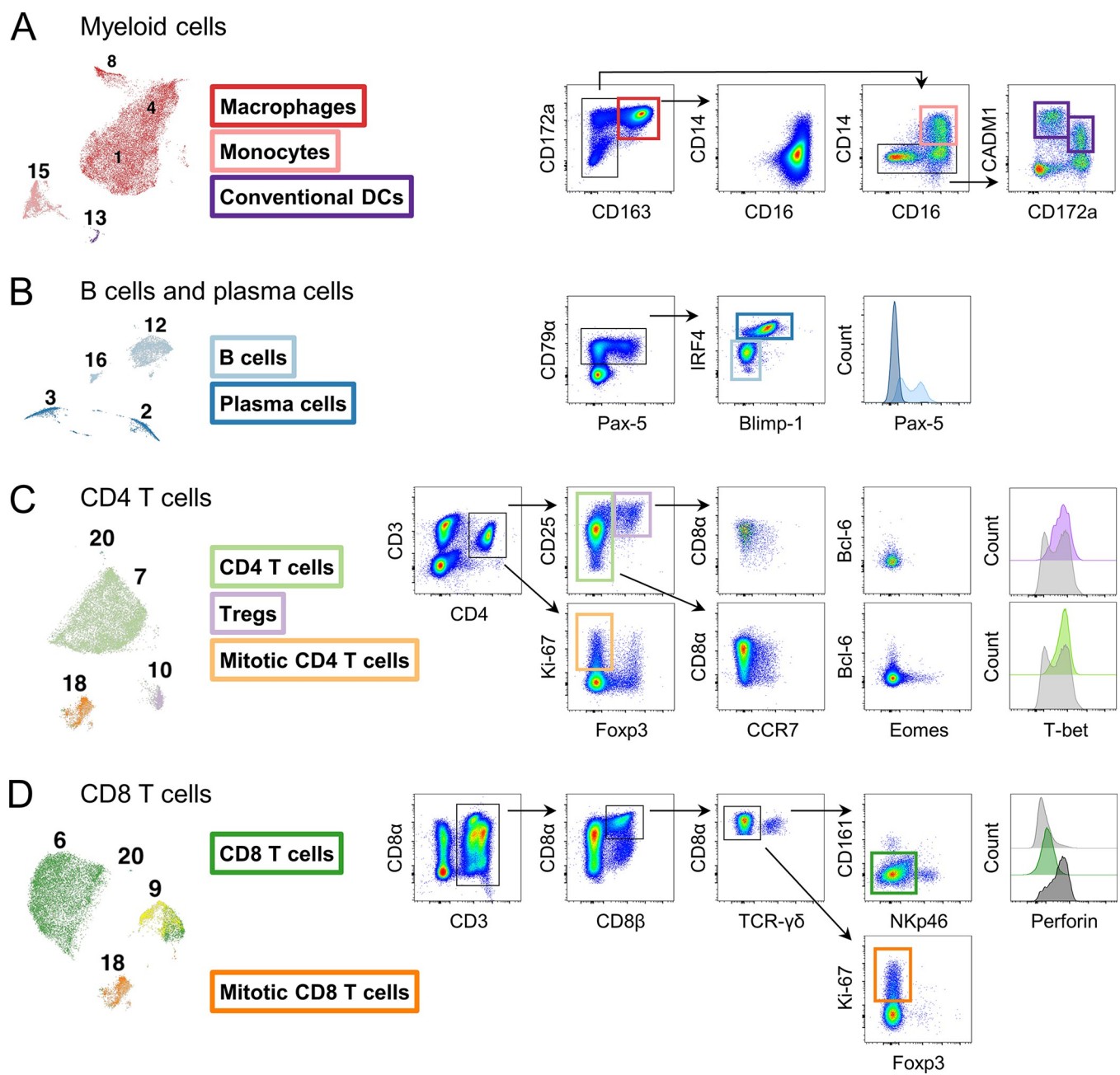

**Fig 2. Confirmation of scRNA-seq cluster annotation by flow cytometry.** Annotation of clusters in scRNA-seq was confirmed by flow cytometry using antibody panels addressing key marker combinations. Colors used in UMAP clustering (left) match with colored gates and histograms in flow cytometry to indicate the same cell type (right). (A) Different subsets of myeloid cells. (B) B cell and plasma cell subsets. (C) CD4 T cell subsets. (D) CD8 T cell subsets.

which may represent "mitotic" CD8 T cells found in cluster 18a. Our sorting strategy for scRNA-seq excluded most NK and γδ T cells present in BAL. However, a flow cytometry panel designed for the characterization of nonconventional T cells (**S2 Table**) indicated the presence of a substantial proportion of γδ T cells, which were largely CD8α$^+$CD16$^-$NKp46$^-$ (**S2C Fig**). Within the CD3$^-$ population, CD161/CD8α defined putative NK cell subsets, which exhibited substantial phenotypic heterogeneity in terms of CD16, NKp46 and perforin expression. This

phenotypic analysis confirmed cell types identified by scRNA-seq and indicated further heterogeneity within NK cells and non-conventional T cells.

## Comparison with PBMCs delineates similar and distinct cell-types

To validate our cell-type annotations and to further examine features of BAL tissue resident populations, we compared our combined treated and control pig BAL scRNA-seq dataset with a publicly available pig PBMC scRNA-seq dataset, published by Herrera-Uribe et al [23]. Both datasets were restricted to shared genes, batch corrected and merged using a mutual nearest neighbors approach (Fig 3). When plotted as a UMAP, several cell types such as monocytes, cDCs, plasma cells (antibody secreting cells, ASCs in the PBMC dataset), B cells and some T cells formed visually integrated and/or adjacent populations between BAL and PBMC (Fig 3A and 3B), implying similar patterns of overall gene expression. Cell-types without near neighbors between the datasets, i.e. BAL macrophages, the BAL 'T cell like' cluster 14 and PBMC $\gamma\delta$ T cells, formed visually distant and distinct populations (Fig 3A and 3B). Looking more closely at T cell populations (Fig 3C) revealed two that partially overlap: BAL conventional T cells with PBMC $\alpha\beta$ T cells (both CD4$^+$ and CD8$^+$) and BAL mitotic T cells with a separate population of PBMC $\alpha\beta$ T cells.

To further delineate how individual cell types compared between BAL and PBMC, the top 20 marker genes for each cluster in the PBMC dataset were used to generate an index for projecting the annotated BAL cells onto the PBMC clusters. Fig 3D provides a visual representation of this mapping, identifying the proportion of cells in each BAL cluster that showed gene expression similarity to each PBMC cluster. To examine which genes were driving or inhibiting mapping between clusters, all clusters and all genes used for mapping were plotted as a heatmap (S3A Fig). Subsets of B cells and T cells were separately plotted (Fig 3E and 3F) using gene clusters identified in S3A Fig alongside defining genes from S1 Table; additional subset heatmaps are also provided in S3B–S3G Fig. Broadly, cell-types in the BAL mapped to the same or closest equivalent cell types in the PBMC dataset. BAL macrophages and monocytes mapped to PBMC monocytes (there are no macrophages in PBMC), BAL cDCs to PBMC cDCs, BAL plasma cells to PBMC ASCs and BAL T cells to PBMC T cells. In some cases, mapping at the subset level was ambivalent. For example, the BAL macrophage clusters mapped to PBMC monocyte clusters 20 and 27 (Fig 3D), suggesting minimal differences in the patterns of top marker gene expression between these PBMC monocyte clusters. At the same time, this example also demonstrated preferential mapping such that BAL macrophages mapped to PBMC monocyte clusters 20 and 27 but not to monocyte clusters 13, 19 or 25. Such preferential mapping was apparent across most cell types analyzed, with only a subset of PBMC clusters identifying as close matches for BAL cells. This implies that the heterogeneity in gene expression that drove the clustering in the PBMC dataset also embodied some of the divergence between BAL and PBMC cell types. Of the 33 PBMC clusters, BAL cells only substantially mapped to 12, suggesting that many of the PBMC cell types/states are distinct from those found in the BAL or relate to cell-types not included in our sorting strategy.

There are also exceptions to expected mappings. In the BAL, both clusters 12 and 16 were annotated as B cells, but only cluster 12 mapped to PBMC B cells, while cluster 16 failed to map (Fig 3D). Fig 3E illustrates the gene expression of these B cell clusters, with BAL cluster 16 uniquely showing reduced expression of numerous ribosomal genes (RPL*, RPS*). Likewise, BAL T cells mapped to PBMC T cells, but there was heterogeneity in mapping of T cell subsets. BAL clusters 18 and 20 were previously subclustered into separate CD4$^+$ and CD8$^+$ populations. However, BAL mitotic T cells (cluster 18) mapped primarily to PBMC CD8$^+$ $\alpha\beta$ T cell cluster 14, and BAL cluster 20 mapped primarily to PBMC CD4$^+$ $\alpha\beta$ T cell cluster 0

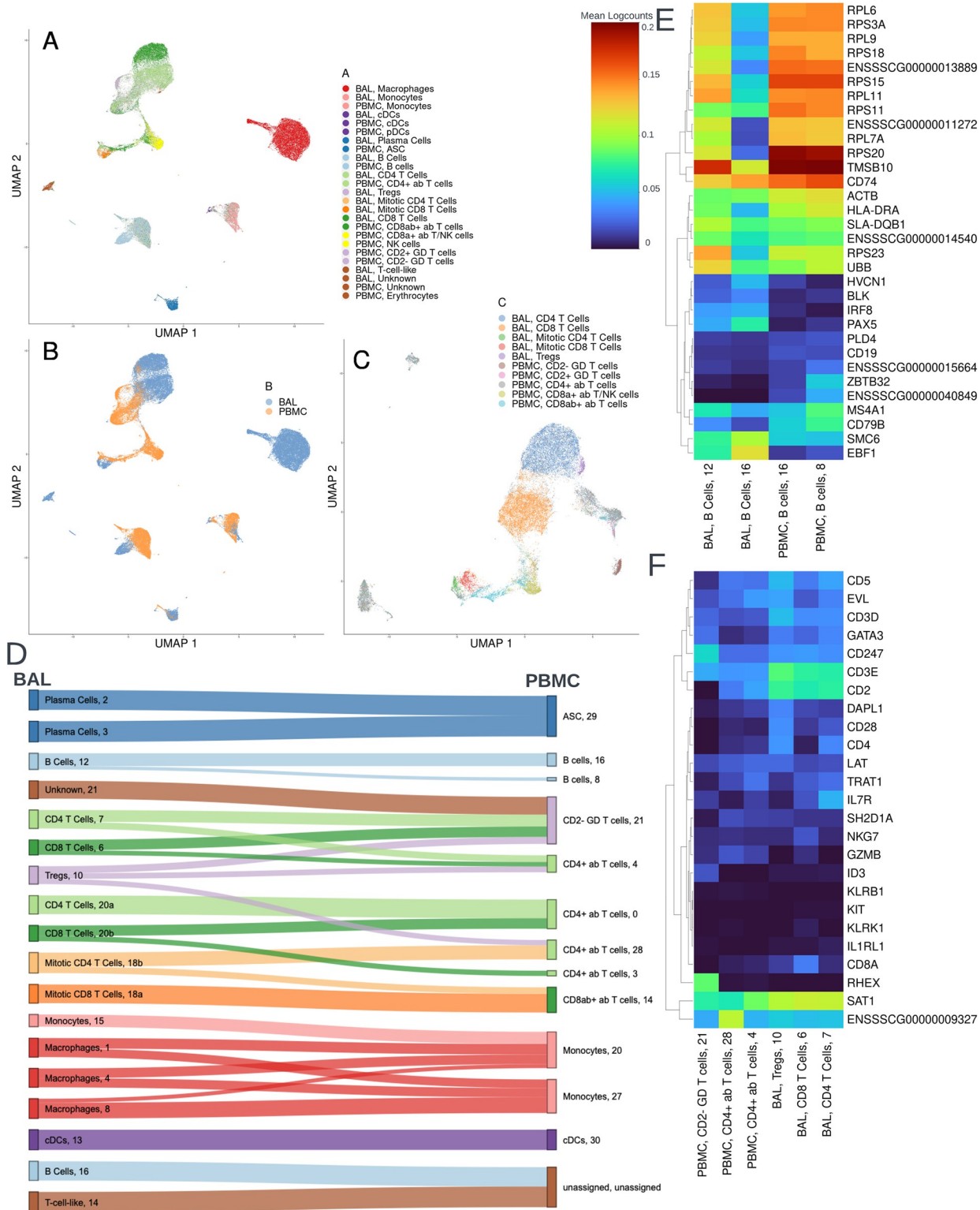

**Fig 3. Comparison of pig BAL and PBMC scRNA-seq datasets.** (A-C) UMAP of BAL and PBMC cells. (A) All cells from BAL and PBMC scRNA-seq datasets colored by cell type. (B) All cells from BAL and PBMC scRNA-seq datasets, colored by tissue. (C) T cells from BAL and PBMC scRNA-seq datasets, colored by cell type. (D) Mapping of BAL cells (left) to PBMC clusters (right) via scMAP, colored by cell type. Each node represents one cluster and is labelled with the cell type and cluster number. Unmapped PBMC nodes and links constituting less than 15% of the cells in a single BAL cluster were omitted for legibility. (E-F) Heatmaps of BAL and PBMC cluster gene expression. Each column represents

one cluster, labelled with the tissue of origin, cell type and cluster number. Color scale is generated from mean logcounts. (E) Heatmap of B cell clusters and (F) Heatmap of T cell clusters from BAL and PBMC scRNA-seq datasets.

(**Fig 3D**). For the largest conventional BAL T cell clusters (6 and 7) the primary mapping was to PBMC CD2$^-$ γδ T cell cluster 21 and, to a lesser extent, PBMC CD4$^+$ αβ T cell cluster 4 (**Fig 3D and 3F**). Furthermore, the 'unknown' BAL cluster 21 also mapped strongly to PBMC CD2$^-$ γδ T cell cluster 21. Conceptually it is unsurprising that a TRM-dominant tissue like the BAL would exhibit distinct gene expression from the circulating T cells that inhabit the peripheral blood. These examples also demonstrate how gene expression representing cellular functions such as cell cycle regulators or cytokine regulation (**Figs 3F and S3**) can outweigh the relatively stable cell surface markers commonly used to distinguish different cell types when performing mapping between datasets using the whole transcriptome.

## Frequencies of Treg cells

We next tested the scRNA-seq data for differential abundance of cells within the annotated cell types between treatment conditions (**S3 Table**). This analysis identified that pigs immunized with Ad-HA/NP+Ad-IL-1β have greatly reduced proportions of cells in cluster 10, annotated as Tregs, within the BAL. Among CD4$^+$ T cells, Tregs make up an average of 0.02% in Ad-HA/NP+Ad-IL-1β immunized pigs versus an average of 0.09% in the other conditions (adjusted p = 7.67e$^{-8}$, **Fig 4A**). This apparent loss of Tregs was not mirrored by any of the other CD4 T cell clusters and was consistent between samples within each condition (**S4A Fig**).

Based on this significant reduction of Tregs in the BAL of IL-1β treated animals seen by scRNA-seq, we further analysed Treg frequency by flow cytometry in BAL and tracheobronchial lymph nodes from the same pigs, plus two additional pigs from immunized and infected groups. Interestingly, there was a significant reduction of Tregs, defined as CD3$^+$CD4$^+$CD25$^{high}$FOXP3$^+$ in the tracheobronchial lymph nodes of the Ad-HA/NP+Ad-IL-1β immunized group compared to the other three groups (**Figs 4B and S5**). However, the reduction in Tregs that was observed in BAL scRNA-seq data was not replicated by the BAL flow cytometry analysis. This is likely because the Treg populations isolated by scRNA-seq and by flow are, whilst overlapping, not the same. In both scRNA-seq and flow cytometry, Tregs are defined as CD3$^+$CD4$^+$FOXP3$^+$ cells, with the flow cytometry analysis also including high protein expression levels of CD25. However, in the scRNA-seq analysis, cells are grouped into clusters based on their top 5000 highly variable genes (HVGs) prior to being identified according to their expression of defining genes. Hence scRNA-seq Tregs (cluster 10, 799 cells), represent a population of T cells that are phenotypically distinct from the other clusters based on 5000 HVGs, and are characterised by higher expression of CD3, CD4, FOXP3 and IL2RA (encoding CD25), among other genes. Cells expressing FOXP3 and IL2RA transcripts also exist in other clusters, including CD4 T cell clusters 7 and 18b (**S4B Fig**). Assuming that transcript levels translate into protein expression, the flow cytometry analysis would include some of these FOXP3 and IL2RA expressing cells alongside cluster 10 as a single Treg population. Hence scRNA-seq Tregs and flow cytometry Tregs represent two overlapping but not identical populations. These data therefore imply that the BAL Treg population that is diminished during IL-1β immunization is a phenotypically distinct subset of BAL Treg and that others are not diminished.

We previously showed that IL-1β is a potent mucosal adjuvant and that intranasal immunization with Ad-HA/NP+Ad-IL-1β increased T cell cytokine production in Babraham pigs [44]. Based on our observations of altered Treg abundance in this condition, we next investigated whether Tregs modulate cytokine production in the BAL. To this end, sorted

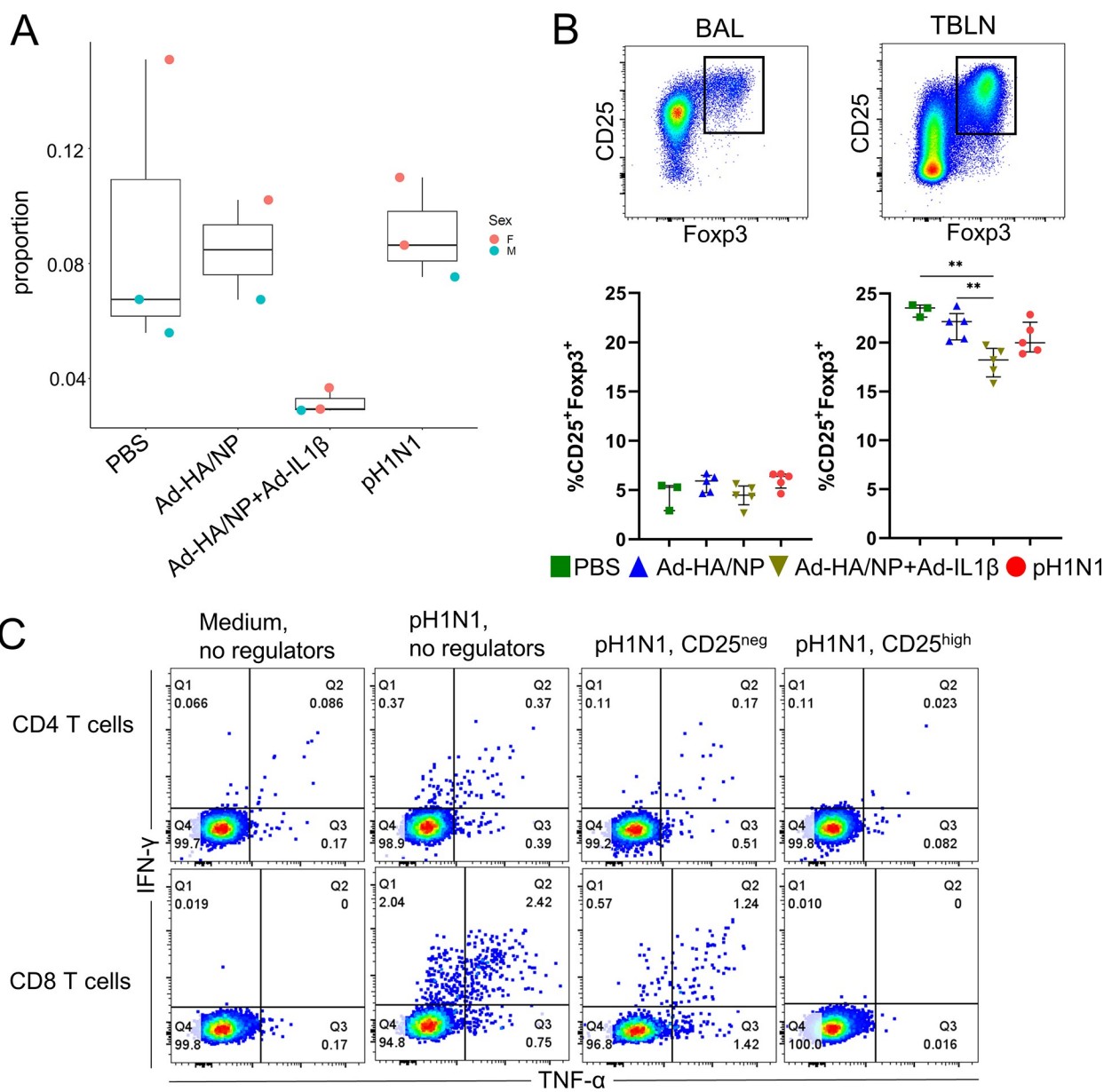

**Fig 4. Abundance and test of suppressive function of Tregs.** (A) Tregs identified by scRNA-seq (cluster 10) as a proportion of all CD4 T cells for each treatment (data from 11 pigs: 5 males and 6 females). (B) Tregs identified by flow cytometry as percentage within CD3$^+$CD4$^+$ T cells for each treatment in cell preparations from BAL and tracheobronchial lymph nodes (TBLN) (** denotes p≤0.01). (C) Suppression of IFNγ and TNFα production in pH1N1 re-stimulated BAL CD4 (top panel) and CD8 T cells (bottom panel) from Ad-HA/NP+Ad-IL-1β immunized pigs, in the presence or absence of CD4$^+$CD25$^{neg}$ or CD4$^+$CD25$^{high}$ sorted BAL cells from Ad-HA/NP immunized pigs. Numbers in quadrants show percentages of cells with the respective phenotype.

CD4$^+$CD25$^{high}$ Tregs and CD4$^+$CD25$^{neg}$ control cells from the BAL of Ad-HA/NP immunized Babraham pigs were co-cultured with BAL cells from Ad-HA/NP+Ad-IL-1β immunized animals in the presence of pH1N1. The expression of IFNγ and TNFα was then assayed by intracellular staining. Incubation of BAL cells derived from Ad-HA/NP+Ad-IL-1β immunized pigs with Tregs abolished IFNγ and TNFα production, while a minor effect was detected when CD4$^+$CD25$^{neg}$ control cells were introduced (**Fig 4C**). Together with scRNA-seq and flow cytometry quantification of Treg populations, these data suggest that IL-1β reduces the

proportion of functionally active Tregs, contributing to increased cytokine production, in agreement with previous studies demonstrating the suppressive role of porcine Tregs [48]. Indeed, the IL-1β treated animals showed increased lung pathology following H3N2 infection [44]. Our data suggests that this is most likely due to the reduced Treg population, and associated enhanced cytokine production, in this group.

## Differential expression analysis reveals condition- and cell type-specific phenotypes

To investigate differential gene expression between our experimental conditions we used a NEgative Binomial mixed model Using a Large-sample Approximation (NEBULA) [49] run discretely on all cells, B cells (clusters 12 and 16), CD4 T cells (clusters 7, 10, 18b and 20a), CD8 T cells (clusters 6, 18a and 20b) and macrophages (clusters 1, 4 and 8) (**S6 Fig, S6 Table**). To distinguish if differentially expressed (DE) genes were distinct or shared between cell types for each experimental condition, significantly DE genes with false discovery rate less than 0.01 were visualized as Venn diagrams (**Fig 5A–5C**). DE genes were also tested for Gene Ontology (GO) enrichment to approximate biological function (**Fig 5F and 5G, S4 Table**). For all experimental conditions, most DE genes were distinct to specific cell types, with only a minority being shared (**Fig 5A–5C**). For Ad-HA/NP immunization versus PBS (**Fig 5A**), macrophages, CD8 T cells and B cells showed significant DE genes, none of which were shared (**Fig 5A**). GO enrichment for Ad-HA/NP immunization versus PBS DE genes returned fewer than 5 DE genes per GO term and is therefore not shown. Ad-HA/NP+Ad-IL-1β immunization versus PBS identified considerably more DE genes and significant GO terms, with 5 DE genes shared between cell types (**Fig 5B and 5D**). DE genes in CD8 T cell populations were enriched for lymphocyte activation and apoptosis-related GO terms, while enrichment testing in macrophages returned only broad GO terms, and analyses in B cells and CD4 T cells returned fewer than 5 DE genes per GO term (**Fig 5G**). The enriched GO terms in CD8 T cells implied that immunization with Ad-HA/NP+Ad-IL-1β elicited a strong immunogenic response, potentially including activation-induced death given the presence of apoptotic terms. Comparing pH1N1 infection with PBS, in addition to numerous distinctly expressed DE genes, NEBULA identified 10 DE genes shared by cell-types **Fig 5C and 5E**). The Interferon Alpha Inducible Protein 6 (IFI6) gene was the only differentially expressed gene to be upregulated in all four cell types. IFI6 is one of many genes induced by interferons and plays a critical role in regulating apoptosis and early control of virus replication [50–54]. GO-term enrichment of DE genes in macrophages presented a trend towards anti-viral response mechanisms, whereas B cells, CD4 T cells and CD8 T cells returned fewer than 5 DE genes per GO term (**Fig 5F**).

Overall, these comparisons suggest that both Ad-HA/NP+Ad-IL-1β immunization and pH1N1 infection invoke a greater immune response than Ad-HA/NP, resulting in greater differential gene expression. These trends and widespread upregulation of IFI6 across BAL cells in pH1N1-infected animals were additionally confirmed using a pseudobulk differential expression analysis.

## IFI6 marks lasting impacts of pH1N1 infection on BAL leukocytes

The differential expression analyses identified IFI6 as strongly upregulated across multiple cell types in pH1N1 infected pigs. Further investigation revealed that IFI6 was upregulated in most scRNA-seq cell clusters in pH1N1 infected pigs (**Fig 6A**). To assess if IFI6 expression was associated with the expression of other genes in different cell types, we used the mixed model of NEBULA to perform a separate differential expression analysis with IFI6 expression itself as the dependent variable, asking which genes were associated with IFI6 expression. Genes

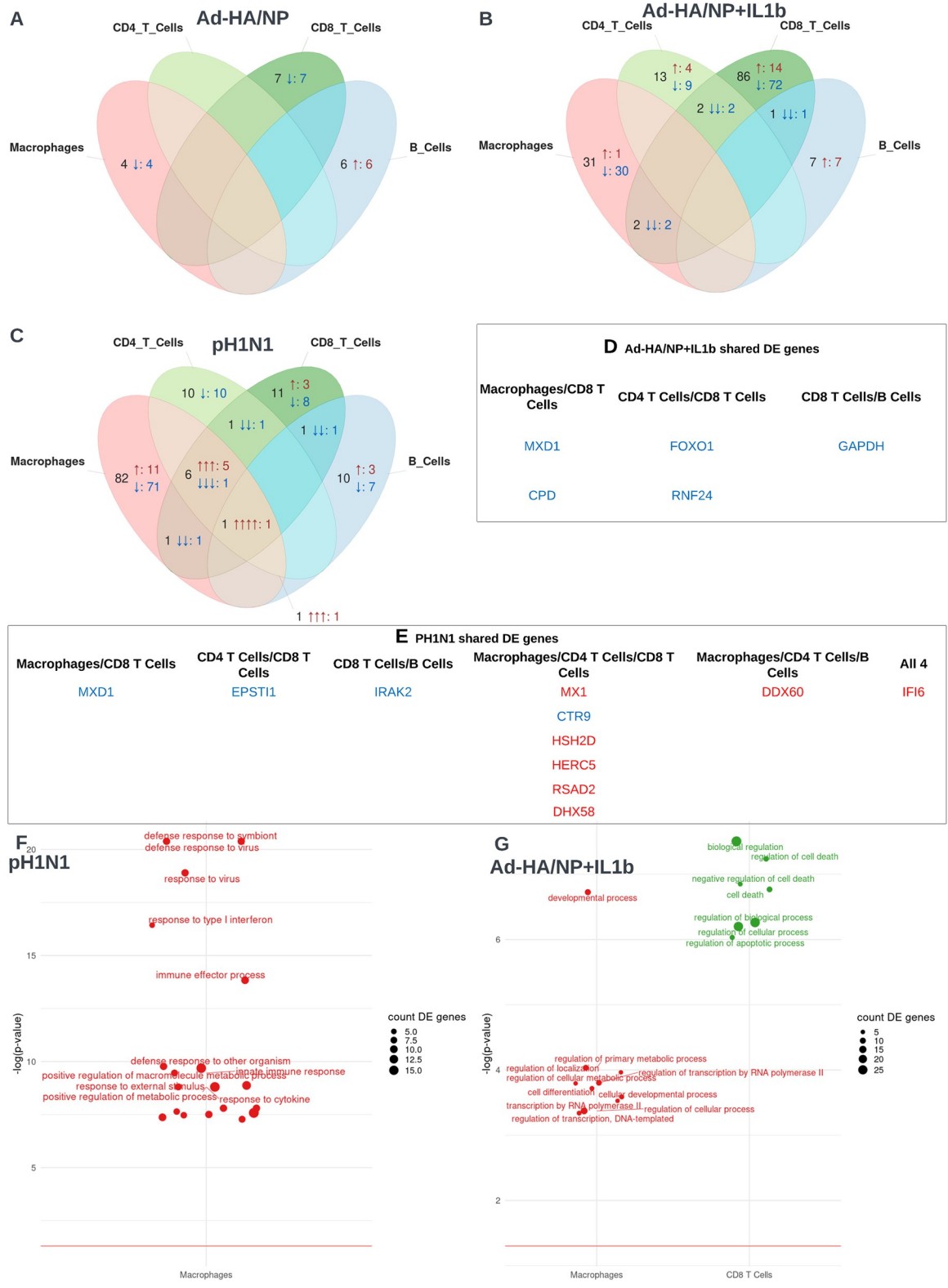

**Fig 5. Differential gene expression with immune challenge** (A-C) Venn diagrams presenting the number of overlapping and distinct differentially expressed genes for macrophages (clusters 1, 4 and 8), CD4[+] T cells (clusters 7, 10, 18b and 20a), CD8[+] T cells (clusters 6, 18a and 20b) and B cells (clusters 12 and 16) under each experimental condition. Numerals indicate the number of differentially expressed genes, and the number of arrows corresponds to the number of cell types in which the genes are differentially expressed. Red

up arrows indicate gene upregulation, blue down arrows indicate gene downregulation. All genes shown have an adjusted p value (Benjamini-Hochberg) of less than 0.01 when comparing gene expression in each condition versus PBS. (D-E) Tables of all differentially expressed genes shared between cell types for each condition. A full table including both shared and unshared genes is available in S4 Table. (F-G) Bubbleplots presenting GO term enrichment for macrophages, B cells, CD4 T cells and CD8 T cells under each experimental condition, based on all genes with an adjusted p value of less than 0.01. GO terms annotated with fewer than five differentially expressed genes have been omitted. (A) Ad-HA/NP treated pigs versus PBS, (B, D, G) Ad-HA/NP+IL-1β treated pigs versus PBS and (C, E, F) pH1N1 infected pigs versus PBS.

associated with IFI6 expression in each cell type were compared in **Figs 6B** and **S7A–S7D**, identifying numerous IFI6 related genes, predominantly in macrophages, CD4 T cells and B cells. GO enrichment (**Fig 6C**) of these genes in CD4 T cells and B cells identified generic immune response terms. For macrophages, IFI6-associated genes were enriched for metabolism-related GO terms.

We also compared the antiviral state of porcine BAL cells between pH1N1-infected and control pigs through a functional interferon assay. To this end, we infected BAL cells with recombinant virus-like particles of vesicular stomatitis virus in which glycoprotein had been replaced with GFP (VSVΔG-GFP) [55, 56]. The kinetics of GFP expression from VSVΔG-GFP was used as a measure of virus replication. The mean GFP expression observed in BAL cells from pH1N1 infected pigs was significantly lower than that in BAL cells from the control group, confirming the antiviral state of BAL cells 21 days post infection with pH1N1 (**Fig 6D** and **S7E–S7F**). This analysis indicates that although it would be expected that cells in BAL always represent a population that had migrated to this site in response to an antigenic stimulus in the respiratory tract, local factors modify gene expression of BAL cells, as demonstrated by the ubiquitous expression of IFI6 following influenza infection.

## Discussion

Studies in animal models have highlighted the unique nature of local lung immunity and the critical role this compartmentalization plays in facilitating protective immune responses to respiratory pathogens. Here we present for the first time a single-cell transcriptomic map of steady-state airways and airways following influenza infection or respiratory immunization in the pig, a highly relevant large natural host model for influenza. We performed our study in inbred Babraham pigs, which have identical MHC (swine leucocyte antigen) and allow for fine-grained immune response analysis [57]. The similar size and anatomy of pig and human organs make this model particularly valuable for translational research.

In pigs, the overwhelming majority of BAL cells exhibit the characteristics of tissue residency. This was shown by intravenous CD3 administration, which will only bind to circulating cells and is the most commonly used method to define tissue residency [45,46]. BAL is thus a particularly useful source of information about tissue-resident immune responses as it reflects the site of respiratory infections and is accessible during bronchoscopy in humans. Single-cell RNA-seq studies of BAL have become more common in recent years, particularly since the COVID-19 pandemic revealed the utility of monitoring the immune response at this accessible site of infection. Studies in humans [18,58], ferrets [19], horses [20], dogs [21] and mice [59] have characterized the various immune and epithelial cells from the BAL, revealing extensive heterogeneity among these populations, substantial differences in their abundances as compared with PBMC [58], and respiratory infection/inflammation-associated changes [18,19,59].

Our study examined BAL extracted from the lung 21 days after vaccine or influenza challenge to understand the lasting impacts of induced immune responses. Irrespective of treatment, porcine BAL cells were dominated by macrophages. However, our sorting strategy allowed us to make key observations about relatively rare lymphocyte populations (e.g. Treg)

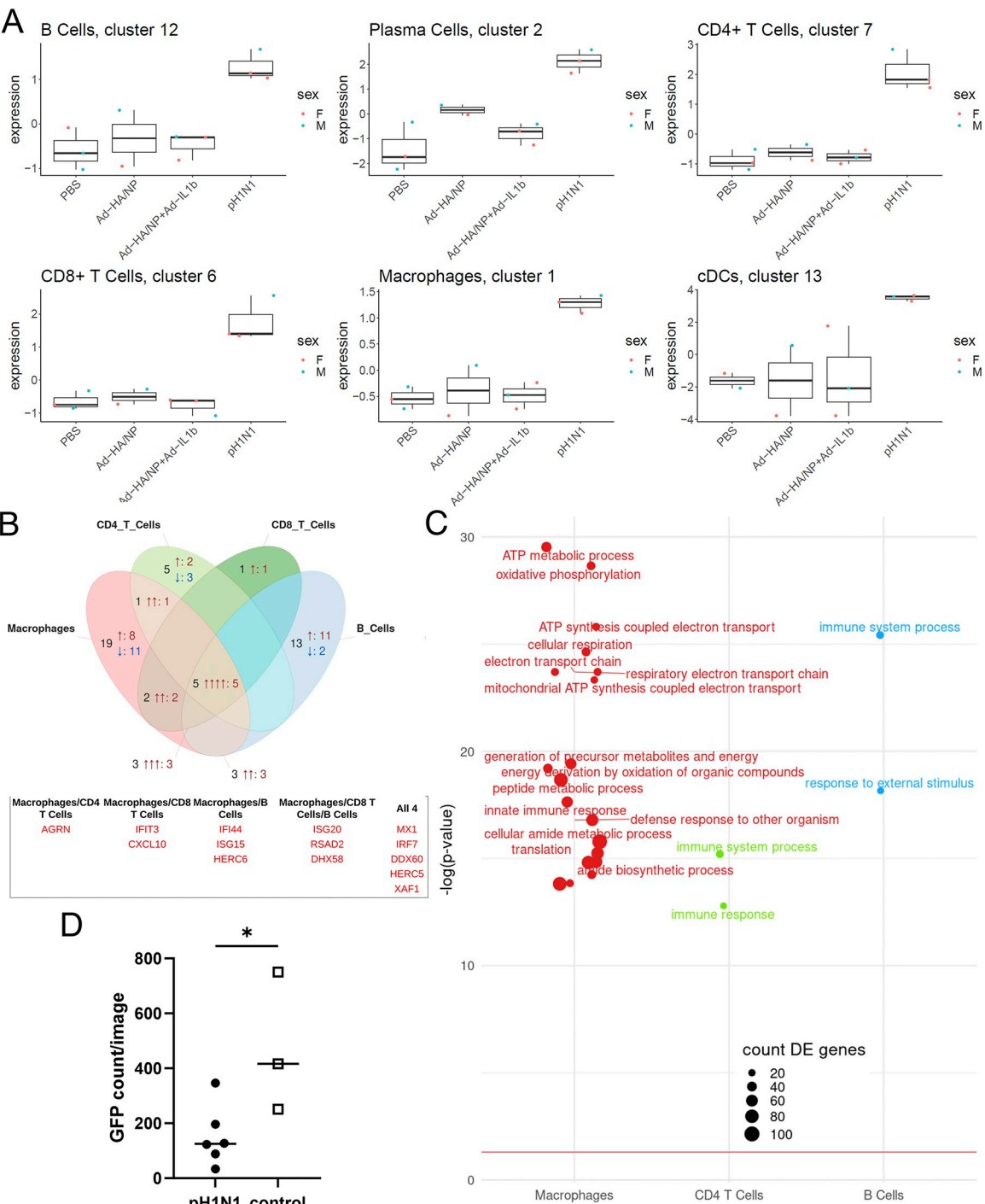

**Fig 6. IFI6 abundance, associated genes and functional consequences of IFI6 expression.** (A) IFI6 expression (normalized log counts per million) from scRNA-seq data across clusters in each experimental condition (data from 11 pigs: 5 males and 6 females). (B) Venn diagram of genes whose expression is associated with IFI6 expression in macrophages (clusters 1, 4 and 8), CD4 T cells (clusters 7, 10, 18b and 20a), CD8 T cells (clusters 6, 18a and 20b) and B cells (clusters 12 and 16). Numerals indicate the number of differentially expressed genes, and the number of arrows corresponds to the number of cell types in which the genes are differentially expressed. Red up arrows indicate gene upregulation, blue down arrows indicate gene downregulation. All genes shown have an adjusted p value (Benjamini-Hochberg) of less than 0.05 and a logFC that is either more than 0.3 or less than –0.3. Accompanying table displays the genes names for all significant genes shared between cell types. A full table including both shared and unshared genes is available in S4 Table. (C) Bubbleplot presenting GO term enrichment for genes differentially expressed as IFI6 is upregulated for macrophages, CD4 T cells and B cells. CD8 T cells have been omitted due to a lack of significant genes. Based on all genes with an adjusted p value of less than 0.05.

GO terms annotated with fewer than five differentially expressed genes have been omitted. (D) GFP counts of BAL cells following infection with VSVΔG-GFP 7 hrs post-infection. Comparison of BAL cells from pH1N1 and control pigs. Asterisk indicates significant difference (p<0.05).

while still capturing information about the phenotype and abundance of more prominent macrophage populations and finding shared gene expression changes across immune cell subsets. Leveraging available porcine PBMC single-cell RNAseq data [23] enabled validation of our cell type annotations and highlighted likely phenotypic differences between tissue and circulating immune cell populations, particularly among T and B lymphocytes.

Our use of frozen BAL samples from a previous immunization study [44] allowed us to gain valuable additional information from a large animal immunological experiment. Comparison between fresh and frozen BAL cells did not reveal significant differences between proportions of total macrophages, T and B cells. While it is possible that particular cell subsets may be affected by liquid nitrogen storage, these effects should be shared across samples making the data comparable if not fully comprehensive. Successful transcriptional profiling of these cells corroborates conclusions drawn in a recent equine study [20] that frozen BAL is a useful source of single-cell transcriptomic information for large-animal model immune responses, opening the door to further re-use of samples from key respiratory immunology studies. By utilizing samples from influenza infected and immunized animals, we had the opportunity to investigate parameters that change under these circumstances. Our data indicate that IL-1β administration resulted in a significant reduction of Treg in the BAL as determined by scRNA-seq compared to the other treatments in our study, although there were no changes in Treg frequency in BAL when identified as $CD3^+CD4^+$, $CD25^{high}Foxp3^+$ at the protein level by flow cytometry in the same samples. This highlights the power of transcriptomic profiling, allowing higher dimensional clustering of immune cell subsets compared to the limited marker sets used in flow cytometry.

Mucosal production of IL-1β has been shown to attract innate and adaptive immune cells by the induction of cytokines and adhesion molecules and to have a direct proliferative effect on CD4 and CD8 T cells [60–62]. Intranasal delivery of IL-1β, together with HA- and NP-encoding adenoviral vectors (Ad-HA/NP+Ad-IL-1β), significantly increased the immunogenicity and heterotypic protection against the influenza virus in mice [30]. However, in pigs, although IL-1β increased antibody and T cell immune responses, this did not reduce lung pathology or viral load following H3N2 influenza challenge [44]. Tregs are responsible for maintaining immune homeostasis by supressing excessive immune responses [48,63]. Of note, effector T cells in the BAL of Ad-HA/NP+Ad-IL-1β immunized pigs were still susceptible to Treg inhibition when cocultured with Tregs from Ad-HA/NP immunized animals. Hence, reduced numbers of functionally active Tregs can lead to imbalance of cytokine production, and this proinflammatory environment can trigger inflammation and tissue damage. Indeed, outbred pigs immunized with AdHA/NP+Ad-IL-1β showed increased lung pathology following challenge infection [44], confirming that maintaining a balance of Tregs is crucial for the proper functioning of the immune system.

Another interesting observation from this study is that pH1N1 infection drives upregulation of IFN alpha-inducible protein 6 (IFI6) in major BAL cell subsets. The expression of IFI6 is tightly regulated in response to IFNα stimulation and plays a crucial role in restricting viral infections including West Nile Virus, Dengue Virus, Yellow Fever, Hepatitis B and Hepatitis C viruses through interference with viral RNA synthesis and translation [51,54,64]. In contrast, a recent report showed that IFI6 can positively affect virus infection and that IFI6 overexpression results in increased influenza replication in cell cultures and in the murine model [65]. In

our study, we observed reduced GFP expression in VSVΔG-GFP infected BAL cells from pH1N1 infected pigs compared to those from control pigs, confirming a persistent antiviral state following influenza infection and suggesting mediation by differentially expressed gene IFI6. Knockout or suppression/inhibition studies would be required to confirm whether the antiviral state we observed is due to IFI6 or other factors. However, irrespective of the mechanism of action and whether conflicting reports of IFI6 function reflect species differences between mice and pigs, or between VSV and influenza infection, the sustained upregulation of IFI6 in our data suggests that the immune system is still actively engaged in combating the virus or resolving inflammation and tissue damage at day 21 post infection. Our results also indicate that although all tissue resident lymphocytes must have a common gene expression programme enabling them to migrate to and survive in tissue sites, local pathogen-related factors can profoundly modify these programs. These modifications are detectable for a surprisingly long time, 21 days, after infection or immunization.

In summary our findings provide a comprehensive high dimensional cellular reference for porcine airways in different immunological contexts, paving the way for a deeper understanding of respiratory physiology and immune responses in pigs.

## Materials and methods

### Ethics statement

Animal experiments were approved by the ethical review processes of The Pirbright Institute and the Animal and Plant Health Agency (APHA) according to the UK Animals (Scientific Procedures) Act 1986 under project license P47CE0FF2.

### Infection and immunization animal studies

Samples obtained from a previous experiment conducted on inbred Babraham pigs were utilized [44]. Briefly three groups of 5 pigs each, were treated as follows: 1) infected intranasally with $8x10^6$ PFU of A/swine/England/1353/2009 (pH1N1) using a mucosal atomization device (MAD); 2) immunized intranasally with $1x10^9$ particles of recombinant adenoviral vectors expressing NP from H1N1/PR8/34 virus and HA from pH1N1/Texas/05/2009 (Ad-HA/NP) using a MAD and 3) immunized intranasally with a mix of $1x10^9$ particles of Ad-HA/NP and $1x10^9$ particles recombinant adenoviral vector encoding porcine IL-1β (Ad-HA/NP+Ad-IL-1β) using a MAD. Control pigs (n = 3) received PBS intranasally. All vaccines and PBS were delivered at 2ml per nostril using a MAD, which reaches both the upper and lower respiratory tract as previously shown using scintigraphy [66]. The animals were culled three weeks later, and bronchoalveolar lavage (BAL), tracheobronchial lymph nodes and blood collected. Virus shedding and immune responses in respiratory tissues and peripheral blood were described in [44].

### Sorting of BAL cell subsets for scRNA-seq

BAL cells had been isolated as described in [44]. After two washes in PBS, cells were resuspended in freezing medium (90% foetal calf serum, 10% DMSO) and subjected to gradual temperature decline in a Mr. Frosty Freezing Container (ThermoFisher) at -80˚C. Subsequently, cryovials were transferred to liquid nitrogen for long-term storage. Cryopreserved BAL cells were thawed, washed and resuspended in PBS. Cells were incubated with antibodies against CD4 (clone 74-12-4, PerCP-Cy5.5-conjugated, BD Biosciences), CD8β (clone PPT23, FITC-conjugated, Bio-Rad), CD3 (clone PPT3, APC-conjugated, Southern Biotech), CD16 (clone G7, AF647-conjugated, Bio-Rad), CD172a (clone 74-22-15, PE-conjugated, Bio-Rad) and

Fixable Viability Dye eFluor780 (ThermoFisher). After two washes in PBS, cells were sorted on a FACSAria UIII (BD Biosciences), equipped with 405, 488, 561 and 640nm lasers and BD FACSDiva 8 software. After gating on FSC-A$^{dim}$SSC-A$^{dim}$ cells and exclusion of doublets and dead cells, CD4$^+$, CD8$\beta^+$ and CD3$^-$CD16$^-$ (to enrich for B cells) cells were sorted (12,000 cells per population). In addition, after gating on FSC-A$^{bright}$SSC-A$^{bright}$ cells, doublet and dead cell exclusion and gating on CD16$^+$CD172a$^+$ cells, 12,000 macrophages were sorted. Representative phenotyping data is shown in **S1A Fig**. Cells from each fraction were pooled and subjected to partitioning and barcoding on a 10x Genomics Chromium iX Controller.

## scRNA-seq library preparation and sequencing

Single cells were processed with the Chromium iX controller, using the Next GEM chip G, and single cell 3' kit v3.1 library preparation kit (10X Genomics, Pleasanton, CA). An estimated 10,000 cells per BAL sample (total of 12 samples) were targeted for recovery during partitioning. Samples were randomly grouped and run with two libraries per sequencing run on three NextSeq High Output 150 cycle reagent kits (Illumina, San Diego, CA), with 1% PhiX, targeting approximately 200 million reads per sample. Bcl files were demultiplexed using 'cellranger mkfastq' and aligned/counted using 'cellranger count' (cellranger-7.0.0) using default parameters and *Sus scrofa* genome (genome assembly 11.1, Ensembl release 107).

## scRNA-seq QC and pre-processing

Data analysis was run using R (version 4.1.1) and RStudio (build 382). Empty droplets, barcode-swapped droplets and ambient RNA were removed from the unfiltered cellranger output using DropletUtils (1.18.1) [67,68]. Genes with no counts and cells with outlying mitochondrial/nuclear gene log-ratios ($>$ 3 median absolute deviations) were removed. One of twelve samples was excluded as a strong outlier due to low sequencing saturation and excessive detected cell number. All remaining samples were normalized by pooled factors [69] and combined to form the analysis dataset. Batch effects were corrected between sequencing runs using fastmnn (batchelor 1.14.0) implementation of mutual nearest neighbors batch correction [70].

## scRNA-seq clustering and annotation

Shared-nearest-neighbors graph-based clustering was performed on batch-corrected data using clusterCells (bluster 1.8.0) with the top 5000 highly variable genes (HVGs), specifically using Jaccard index to weight edges and the Louvain method for community detection, with a k value of 20. This generated 21 initial clusters. Cluster 5 exhibited particularly low counts, limited numbers of genes detected and lack of any positive defining features and was therefore defined as poor-quality and excluded from further analysis. Co-expression based doublet scoring by cxds (scds 1.14.0) indicated that cluster 17 was likely comprised of a large proportion of doublets. Cluster 11 also had a substantial proportion of doublets, and manual inspection revealed co-expressed macrophage and B cell marker genes, suggesting that this cluster was also likely doublets. These clusters were therefore excluded from further analysis. Clusters 18 and 20 were found to express a dichotomy of CD4 and CD8 expression and so were split into CD4$^+$ and CD8$^+$ subclusters. This was achieved by identifying HVGs within the cluster of interest, finding those that correlated best with CD4 and CD8B expression using a zero-inflated Kendall's Tau correlation metric implemented in scHOT (v1.10.0), and then re-running batch correction and subclustering (Jaccard index to weight edges and the fast greedy modularity optimization algorithm for community detection) on cells in the cluster of interest. Varied expression of genes including CD3E, CD8A, CD8B, RORC and KLRB1 within cluster 9 also prompted sub-clustering. This was done in an unsupervised manner by identifying the

top 2000 HVGs in cluster 9 cells and then re-running batch correction and clustering (Jaccard index to weight edges and the fast greedy modularity optimization algorithm for community detection). This split the population into three subclusters.

Clusters were manually annotated with conventional names (cell-types) by identifying the positive and negative expression of typical defining markers, as listed in **S1 Table** and illustrated in **Figs 1C** and **S1B,** and looking at top marker genes within each cluster as defined by the scoreMarkers function (scran 1.26.0, **S5 Table**).

Cluster 19 showed strong expression of ACTG1, GPR37, NAPSA and SFTPA1, typically expressed by mucus producing epithelial cells. Cluster 9 and its subclusters were enriched for NK cell typical genes such as KLRK1, KLRB1, KIT and NCR1. As our cell sorting strategy was not intended to capture NK cells, NK-cell-like T cells or epithelial cells, clusters 9 and 19 risked presenting a biased representation of these larger cell populations and were therefore removed from further analysis.

## scRNA-seq analysis, differential abundance analyses

To compare the abundance of cells within each cluster across conditions, we performed differential abundance analyses using a negative binomial generalized linear model with empirical Bayes quasi-likelihood F-tests [71,72], as implemented in edgeR (v 3.40.0). We included sex as a covariate and fitted the model with all conditions to test for differential abundance in any condition. The number of cells per cluster was normalized by sample cell number within the comparison, and trend estimation was turned off when estimating dispersion and quasi-likelihood dispersion. To account for the sort and pool strategy used to balance cell types among the sequenced cells, clusters were divided according to their membership of the 4 sorted cell types (macrophages, CD4 T cells, CD8 T cells, B cells) before running differential abundance analyses within each. Those clusters that did not obviously fall into one of these cell types were excluded in order to keep the universes for differential abundance analysis equivalent to or smaller than the sorted cell types, thereby avoiding sorting biases in intra-cell-type differential abundance analyses.

## scRNA-seq analysis, differential expression analyses

Differential expression analyses for macrophages (clusters 1, 4 and 8), CD4 T cells (clusters 7, 10, 18b and 20a), CD8 T cells (clusters 6, 18a and 20b) and B cells (clusters 12 and 16) (**Figs 5** and **6**) were performed using a NEgative Binomial mixed model Using a Large-sample Approximation (NEBULA) [49], using PBS treated BAL as a baseline, blocking on sample and cluster of origin (to account for the effect of differential cell numbers of clusters between treatment groups). Final p values were adjusted using the Benjamini-Hochberg procedure [73] as implemented in p.adjust (stats v4.1.1). All results from this analysis are provided in **S6 Table**. To mitigate against potential issues with limited numbers of subjects [49], we also ran pseudo-bulk differential expression analyses within each cluster. Briefly, for each comparison of immune challenge versus PBS, counts from all cells in each cluster in each sample were pooled, limiting to clusters with at least 10 cells in at least 2 samples from each condition. For each cluster, we then fitted a negative binomial generalised linear model, blocking on sex, and tested for differential expression between conditions using empirical Bayes quasi-likelihood F-tests, as implemented in edgeR (v 3.40.0).

## Gene Ontology Enrichment

Lists of differential expressed genes were passed to topGO (2.44.0) using the Genome wide annotation for pig (DOI: 10.18129/B9.bioc.org.Ss.eg.db, release 3.13.0), biological process

subontology and a minimum node size of 5. Enrichment tests for classicFisheralong with all results are available in **S4 Table**.

## scRNA-seq analysis, comparison with PBMCs

For comparison with pig PBMC transcriptomic characteristics, scRNA-seq data generated by Herrera-Uribe et al. 2021 [23] was downloaded under license CC BY-SA 4.0 from: https://data. nal.usda.gov/dataset/data-reference-transcriptomics-porcine-peripheral-immune-cells-created-through-bulk-and-single-cell-rna-sequencing/resource/7c096891-2807-4e4e-8f5f-0c02c6f434c4

This PBMC dataset was then filtered to include only genes shared between the BAL and PBMC datasets (13050 shared genes); clustering and cell type annotation in the PBMC dataset was left unchanged. The top 20 differentially expressed genes for each cluster within each dataset (55 total clusters) were identified using findMarkers (scran 1.20.1) and combined (402 final genes after removing duplicates) to provide a data subset for mutual nearest neighbors correction [70]. mnnCorrect (batchelor 1.8.1) was used to correct for batch effects in each dataset and then to merge the datasets prior to plotting of joint UMAPs.

To directly compare gene expression between cell types and clusters, the top 20 defining genes for each cluster in the PBMC dataset (36 clusters) were identified using findMarkers (scran 1.20.1), duplicates removed (268 final genes) and used as selected features to index the clusters in the batch corrected PBMC dataset via scmap (scmap 1.14.1). The clusters in the batch corrected BAL dataset were then projected onto the indexed PBMC dataset via scmap, using the default threshold of 0.7 (i.e. 70%, equal to 188 genes), and the results plotted as a Sankey plot using networkD3 (networkD3 0.4).

To identify the sets of genes driving the mapping between BAL and PBMC clusters, all genes and all clusters inputted into scmap were plotted as a heatmap (**S3A Fig**), with clusters of genes manually annotated (guided by Gene Ontology terms and the literature). Selected subsets of cell clusters and gene clusters were then plotted as separate heatmaps for **Figs 3E and 3F** and **S3B–S3G**.

## Flow cytometry of BAL cells

BAL cells were thawed, washed in PBS, plated into 96-well round bottom plates for flow cytometry staining and labelled with antibodies grouped into four staining panels (**S2 Table**). Some antibodies were labelled with isotype or species-specific secondary antibodies conjugated to fluorochromes. Biotinylated antibodies were labelled with streptavidin conjugates. Free binding sites of secondary antibodies were blocked with mouse IgG (1μg per sample, Jackson) prior to incubation with further, directly conjugated antibodies. Live/dead discrimination was achieved by labelling of samples with VDeFluor780 (ThermoFisher). Following labelling of surface markers, cells were fixed and permeabilized with eBioscience Foxp3/ Transcription Factor Staining Buffer Set (ThermoFisher) according to manufacturer's instructions, followed by labelling with antibodies against intracellular targets. Incubation steps for extracellular markers lasted for 20 minutes, incubation with antibodies for intracellular targets was performed for 30 minutes. Cells were kept at 4˚C at all times. Cells from BAL and TBLN were analysed on a Cytek Aurora spectral flow cytometer, equipped with 5 lasers (UV 355nm, Violet 405nm, Blue 488nm, Yellow-Green 561nm, Red 640nm) and 64 fluorescence detection channels UV: 16, Violet: 16, Blue: 14, Yellow-Green: 10, Red: 8. Per sample, at least $2 \times 10^5$ cells were acquired. Data was acquired using Cytek's SpectroFlo software and later analysed with FlowJo version 10 (BD Biosciences).

## Analysis of Treg function

BAL cells from Ad-HA/NP immunized animals were stained with CD4 PerCP-Cy5.5 (as in **S2 Table**) and CD25 (K231.3B2, AF488-conjugated, Bio-Rad). CD4$^+$CD25$^{high}$ and CD4$^+$CD25$^{neg}$ cells were sorted on a FACSAria UIII (**S4C Fig**). To determine the effect of Tregs, 4 x 10$^4$ cells of the sorted cell populations were added to 2 x 10$^6$ BAL cells from animals immunized with Ad-HA/NP+Ad-IL-1β. These cocultures were stimulated with H1N1pdm09 (MOI of 1) or medium control overnight. Intracellular staining was performed to analyse IFNγ and TNFα production by CD4 and CD8 T cells. Brefeldin A (GolgiPlug, BD Biosciences) was added, and 5 h later cells were stained with mAbs against CD4-PerCP-Cy5.5 (as in **S2 Table**) and CD8β-FITC (clone PPT23, Bio-Rad). The cells were fixed and permeabilized with BD Cytofix/Cyto-perm (BD Biosciences) as per the manufacturer's instructions. The cells were then stained with antibodies against IFNγ (P2G10, AF647-conjugated, BD Biosciences) and TNFα (Mab11, BV421-conjugated, BioLegend). Additionally, Near-Infrared Fixable LIVE/DEAD stain (Invitrogen) was used for identification of live cells.

## Virus infectivity assays to determine antiviral state of pig BAL

Pig BAL cells from pH1N1 infected or control groups were challenged with infectious recombinant VLPs of vesicular stomatitis virus (VSV-ΔG-GFP) in which the glycoprotein was replaced with GFP which was used as a marker to monitor the replication of VSV-ΔG-GFP. The procedure for infecting cells with VSV-ΔG-GFP was previously described [55,56,74] with slight modifications in which we measured the kinetics of GFP expression from VSV-ΔG-GFP in a live cell imager (IncuCyte-S3).

## Supporting information

**S1 Fig. scRNA-seq of sorted BAL immune cells, related to Fig 1.** (A) Sorting strategy for balanced analysis of macrophages, CD4 T, CD8 T and B cells. Following exclusion of doublets and dead cells, macrophages were sorted based on CD172a and CD16 co-expression (top panel). CD4 T cells were sorted based on CD4 expression, and CD8 T cells based on expression of CD8β. Due to a lack of a surface pan-B cell marker for pig, B cells were enriched based on a CD3⁻CD16⁻CD172a⁻ phenotype (bottom panel). (B) Heatmap of defining gene expression across all pig BAL scRNA-seq clusters. (C) Cluster occupancy by sample. (D) TSNE plot of pig BAL cells analysed by scRNA-seq.
(PDF)

**S2 Fig. Gating strategy for BAL samples and phenotyping of unconventional T cells, related to Fig 2.** (A) For phenotyping of myeloid cells, FSC-A$^{high}$SSC-A$^{high}$ cells were gated, followed by exclusion of doublets and dead cells (low fluorescence intensity for live/dead dye). (B) As in (A) but for phenotyping of lymphocyte subsets, initially FSC-A$^{low}$SSC-A$^{low}$ cells were gated, followed by exclusion of doublets and dead cells. (C) Unconventional T cells, which had been largely excluded from scRNA-seq by the applied sorting strategy, were investigated by flow cytometry. After exclusion of dead cells and doublets as shown in (B), γδ T cells (CD3$^+$TCR-γδ$^+$), non-γδ T cells (CD3$^+$TCR-γδ$^-$) and non-T cells (CD3$^-$TCR-γδ$^-$) were gated and analysed for expression of CD8α, CD161, CD16, NKp46 (CD335) and perforin.
(PDF)

**S3 Fig. Comparison of pig BAL and PBMC scRNA-seq datasets, related to Fig 3.** (A) Heatmap of all cell clusters and genes used in the BAL to PBMC single-cell transcriptome mapping by scMAP. (B-G) Additional heatmaps of cluster and gene subsets used in the BAL to PBMC

mapping.
(PDF)

**S4 Fig. Treg abundance and sorting of BAL Tregs, related to Fig 4.** (A) Proportions of CD4 T cell clusters (7, 10, 18b, 20a) across all BAL samples in scRNA-seq data. Different treatments are indicated at the bottom of the graph. (B) UMAP plots of all cells from pig BAL scRNA-seq analysis, colored by normalised logcounts of IL2RA (CD25) and FOXP3. (C). Gating hierarchy applied for sorting of CD4$^+$CD25$^-$ (control) and CD4$^+$CD25$^{high}$ (Treg) cells from BAL of Ad-HA/NP treated pigs.
(PDF)

**S5 Fig. Original flow cytometry data for Treg identification, related to Fig 4B.** Following pre-gating on CD3$^+$CD4$^+$ T cells (as in Fig 2C), Treg were identified by CD25/Foxp3 co-expression. (A) BAL samples from all animals in the different treatment groups. (B) As in (A) but samples from TBLN. (A+B) The numbers in top right corners represent the animal identifiers. Percentages of CD25$^+$Foxp3$^+$ cells are given in the lower right corner of each pseudocolor plot.
(PDF)

**S6 Fig. Volcano plots of differentially expressed genes in different pig BAL cell subsets for each experimental condition, related to Fig 5.** (A-C) Macrophages, (A) Ad-HA/NP treated pigs versus PBS, (B) Ad-HA/NP+Ad-IL-1β treated pigs versus PBS and (C) pH1N1 infected pigs versus PBS. (D-F) CD4 T cells, (D) Ad-HA/NP treated pigs versus PBS, (E) Ad-HA/NP +Ad-IL-1β treated pigs versus PBS and (F) pH1N1 infected pigs versus PBS. (G-I) CD8 T cells, (G) Ad-HA/NP treated pigs versus PBS, (H) Ad-HA/NP+Ad-IL-1β treated pigs versus PBS and (I) pH1N1 infected pigs versus PBS. (J-L) B cells, (J) Ad-HA/NP treated pigs versus PBS, (K) Ad-HA/NP+Ad-IL-1β treated pigs versus PBS and (L) pH1N1 infected pigs versus PBS. Horizontal red line is at–$\log_{10}$ (0.05). Vertical red lines are at –0.6 and 0.6. Adjusted p values were calculated using the Benjamini-Hochberg procedure. (A-C) Adjusted p values below $10^{-20}$ were capped at $10^{-20}$. (D-L) Adjusted p values below $10^{-10}$ were capped at $10^{-10}$.
(TIF)

**S7 Fig. Differentially expressed genes associated with IFI6 upregulation and VSV infection of BAL cells, related to Fig 6.** (A-D) Volcano plots of genes whose expression is associated with that of IFI6 in macrophages, CD4 T cells, CD8 T cells and B cells. Horizontal red line is at –$\log_{10}$(0.05). Vertical red lines are at –0.3 and 0.3. Adjusted p values were calculated using the Benjamini-Hochberg procedure. Adjusted p values below $10^{-100}$ were capped at $10^{-100}$. (A) Macrophages. (B) CD4 T cells. (C) CD8 T cells. (D) B cells. (E-F) GFP expression from infectious recombinant VLPs of vesicular stomatitis virus expressing GFP (VSVΔG-GFP) in porcine BAL cells at 7 hrs post-infection captured in live cell imager (IncuCyte-S3). Scale bars: 400nm. (E) GFP expression from VSVΔG-GFP at 7 hrs post-infection in H1N1 infected porcine BAL cells, (F) GFP expression from VSVΔG-GFP at 7 hrs post-infection in naïve (control) porcine BAL cells.
(PDF)

**S1 Table. scRNA-seq cluster and cell type annotation.** Table showing all clusters, their annotated cell type, and the list of HGNC genes used to delineate cell types. Green gene names indicate positive expression relative to other clusters, red indicates negative expression. High (hi) and low (lo) have been used when average gene expression between clusters is less distinct. CD3D(split) and CD3E(split) refers to the fact that cluster 14 includes two populations with

diverging CD3D and CD3E expression.
(PDF)

**S2 Table. Antibodies and second step reagents used for flow cytometry.** Details on antibodies for each staining panel (myeloid cells, unconventional T cells, CD4 and CD8 T cells, B cells and plasma cells) are given. Second-step reagents are indicated by footnotes.
(PDF)

**S3 Table. Differential abundance results.** Results from differential abundance analyses between treatment conditions using a negative binomial generalized linear model with empirical Bayes quasi-likelihood F-tests. Includes tests for each cluster identified as any of; CD4 T cells (CD4), CD8 T cells (CD8), macrophages or B cells and plasma cells (B).
(XLSX)

**S4 Table. Differential expression gene lists and GO enrichment results.** Lists of differentially expressed genes for each treatment condition versus naive, split by cell type, macrophages (clusters 1, 4 and 8), CD4 T cells (clusters 7, 10, 18b and 20a), CD8 T cells (clusters 6, 18a and 20b) and B cells (clusters 12 and 16). Red HGNC gene names suffixed with '-Up' imply significant upregulation in that cell type(s), blue gene names suffixed with '-Down' imply significant downregulation in that cell type(s). Also included are GO enrichment results from topGO, showing the top20 significant (by Fisher's exact test) GO terms for each combination of treatment condition and cell type as tested against naive.
(XLSX)

**S5 Table. Lists of gene expression scores as outputted by scoreMarkers.** Lists of genes and scores as outputted by the scoreMarkers function (scran 1.26.0), ranking genes by their degree of distinctive expression versus all other clusters.
(ZIP)

**S6 Table. NEBULA differential expression results.** Lists of genes and their respective p values as performed using a NEgative Binomial mixed model Using a Large-sample Approximation (NEBULA) for each treatment condition and cell type combination versus naive. Adj.P.Val were calculated using the Benjamini-Hochberg procedure as implemented in p.adjust (stats v4.1.1).
(TXT)

## Acknowledgments

We thank the flow cytometry and bioinformatics facilities at the Pirbright Institute, and Simon Andrews and the Bioinformatics core at the Babraham Institute for enabling this research. We also thank the staff at the Biological Research Facility and Immunological Toolbox (https://www.ed.ac.uk/roslin/facilities-resources/immunological-toolbox) for their assistance in producing the monoclonal antibody to porcine CD161.

## Author Contributions

**Conceptualization:** Wilhelm Gerner, Arianne C. Richard, Elma Tchilian.

**Data curation:** Andrew Muir, Arianne C. Richard.

**Formal analysis:** Andrew Muir, Basudev Paudyal, Selma Schmidt, Nicos Angelopoulos, Graham Freimanis, Arianne C. Richard.

**Funding acquisition:** Matthias Tenbusch, Wilhelm Gerner, Arianne C. Richard, Elma Tchilian.

**Investigation:** Andrew Muir, Basudev Paudyal, Selma Schmidt, Ehsan Sedaghat-Rostami, Soumendu Chakravarti, Sonia Villanueva-Hernández, Katy Moffat, Noemi Polo, Anna Schmidt, Graham Freimanis, Wilhelm Gerner, Arianne C. Richard, Elma Tchilian.

**Methodology:** Andrew Muir, Katy Moffat, Wilhelm Gerner, Arianne C. Richard, Elma Tchilian.

**Project administration:** Wilhelm Gerner, Arianne C. Richard, Elma Tchilian.

**Resources:** Arianne C. Richard, Elma Tchilian.

**Software:** Andrew Muir, Arianne C. Richard.

**Supervision:** Matthias Tenbusch, Wilhelm Gerner, Arianne C. Richard, Elma Tchilian.

**Validation:** Andrew Muir, Selma Schmidt, Noemi Polo, Nicos Angelopoulos, Matthias Tenbusch, Arianne C. Richard.

**Visualization:** Andrew Muir, Selma Schmidt, Ehsan Sedaghat-Rostami, Soumendu Chakravarti, Sonia Villanueva-Hernández, Wilhelm Gerner, Arianne C. Richard.

**Writing – original draft:** Andrew Muir, Wilhelm Gerner, Arianne C. Richard, Elma Tchilian.

**Writing – review & editing:** Andrew Muir, Basudev Paudyal, Selma Schmidt, Ehsan Sedaghat-Rostami, Soumendu Chakravarti, Sonia Villanueva-Hernández, Katy Moffat, Noemi Polo, Nicos Angelopoulos, Anna Schmidt, Matthias Tenbusch, Graham Freimanis, Wilhelm Gerner, Arianne C. Richard, Elma Tchilian.

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
