## [Decision Letter · Decision Letter 0]

30 Mar 2024

Dear Gerner,

Thank you very much for submitting your manuscript "Single-cell analysis reveals lasting immunological consequences of influenza infection and respiratory immunisation in the pig lung" for consideration at PLOS Pathogens. As with all papers reviewed by the journal, your manuscript was reviewed by members of the editorial board and by several independent reviewers. In light of the reviews (below this email), we would like to invite the resubmission of a significantly-revised version that takes into account the reviewers' comments.

Both reviewers thought that the present study contributes novel and interesting data that would be of interest to the field. However, the dense data structure required further explication and/or qualification of the key conclusions. 

We cannot make any decision about publication until we have seen the revised manuscript and your response to the reviewers' comments. Your revised manuscript is also likely to be sent to reviewers for further evaluation.

Sincerely,

Amy L. Hartman, PhD

Academic Editor

PLOS Pathogens

Benhur Lee

Section Editor

PLOS Pathogens

Michael Malim

Editor-in-Chief

PLOS Pathogens

orcid.org/0000-0002-7699-2064

Reviewer's Responses to Questions

**Part I - Summary**

Reviewer #1: The manuscript from Muir et al. “Single-cell analysis reveals lasting immunological consequences of influenza infection and respiratory immunisation in the pig lung” builds on prior work by the same group that tested an adenoviral vectored vaccine expressing hemagglutinin and nucleoprotein in the presence or absence of IL-1β against influenza in pigs, which resulted in significantly increases antibody and T cell responses but not increase protection. In this study, the authors used stored bronchoalveolar lavage samples from that previous swine study from pigs vaccinated with the Ad-HA/NP vaccine, the Ad-HA/NP+Ad-IL-1β or challenged with a pandemic H1N1 virus. Samples were analyzed by single cell RNA sequencing (scRNAseq) to characterize leukocyte populations and compared to flow cytometry. Data obtained here were validated and compared to previously available data from swine PBMCs. Their scRNAseq sorting strategy was able to detect the largely dominating macrophage population as well as less abundant populations, including more rare Treg populations. The data indicates substantial reduction in the Treg population in pigs immunized with the vaccine containing IL-1β compared to other treatments, which resulted in increased IFNγ and TNFα expression. Further, they observed that infection with pH1N1 infection led to upregulation of IFN alpha-inducible protein 6 (IFI6) even 21 days post infection, likely involved in a prolonged antiviral state even weeks after infection. The data broadens understanding about the immune responses in swine airways in response to different immunological stimulations, adding valuable knowledge to the immunology field. However, the manuscript has a focus on the immunology context per se and lacks discussion on the response to the pathogen or host-pathogen interaction, and may be better suited for an immunology specific journal.

Reviewer #2: The authors used single RNA-sequencing (scRNA-seq) and flow cytometry to characterize the major leucocyte subsets in frozen bronchoalveolar lavage (BAL). They compared BAL cells collected 21 days after H1N1pdm09 infection or respiratory immunization with an adenoviral vector vaccine expressing HA and NP proteins (Ad-HA/NP) with or without IL-1β to BAL from PBS treated control pigs. They compared their scRNA-seq BAL cluster maps to peripheral blood cells highlighting differences between tissue resident and circulating immune cells. They conclude: 1) mucosal administration of IL-1β reduced the number of functionally active Treg; 2) flu infection upregulated IFI6 in BAL cells, thus decreasing their susceptibility to virus replication in vitro. They note that some vaccines fail despite inducing powerful immune responses and that caution is needed when applying results from small animals like mice to humans, and thus the importance of the pig as a model to study disease in humans and livestock.

This is an interesting and valuable evaluation of frozen pig BAL cell subsets collected from influenza infected or vaccinated pigs, leading to identifying potential response controlling cell subsets.

The authors need to improve their data presentation and clarify exact methods as noted below.

**Part II – Major Issues: Key Experiments Required for Acceptance**

Reviewer #1: The flow cytometry analysis revealed significant differences in cell population phenotype compared to scRNAseq, particularly Treg. Could this be a difference in sensitivity for the assays? That was not discussed. Also, cryopreservation has been shown to affect cell viability and binding of markers, and could be the factor resulting in differences observed. There was no information about the cryopreservation method. Please explain how the BAL was frozen and if there were any cryopreservation methods used. There should be discussion on how the freezing process could affect cell populations.

Data here shows that IL1beta downregulates Tregs even when compared to PBS. This could lead to a cytokine storm and more severe disease upon subsequent infection. The authors mention this quickly for their previous work data, but this should be discussed further when discussing results in Fig 4.

Reviewer #2: 1. The BAL samples were collected from 18 pigs, of which 3 from each treatment group were used for their studies. However, the authors do not clearly state in the beginning of Results, or in their M&M, whether BAL cells were pooled for their summary analyses or if the PBS treatment group was used for the basic analyses (Figs1-3). Only on l.280 do they start noting comparison of data for different treatment conditions.

2. L342 “IFI6 was particularly notable as being universally upregulated across leukocyte subsets.” How did the authors identify just IFI6? How many other genes were expressed across cell subsets? More data must be presented to validate the singular role of IFI6.

3. Fig,6 legend (A) IFI6 expression across clusters in each experimental condition. What actual data is this? Please clarify. How many data points for each graph? How many pigs – the 18 original? What’s the overall sex distribution?

**Part III – Minor Issues: Editorial and Data Presentation Modifications**

Reviewer #1: L34: single cell RNA sequencing

L314-316: “The addition of Tregs (CD4+CD25high) to the BAL nearly abolished cytokine production, while a minor effect was detected when CD4+CD25neg control cells were introduced (Figure 4C).” this sentence is a little confusing, not clear what was done since this comes prior to method. Perhaps “the incubation of Tregs with BAL cells of Ad-HA/NP+Ad-IL-1β immunized pigs nearly abolished IFNγ and TNFα production…” is a clearer way to describe results.

L341: This is the first time IFI6 is mentioned in the main text. Please spell out and explain a little more of the function of IFI6 to audience who is not expert in IFN response.

L377: How is it pulmonary immunization done? I thought this was intranasal.

L382-383: Here and in many other instances, it is not clear if results mentioned are for controls, or infected, or vaccinated. That should always be specified. If referring to response in general, then state "in all treatment groups" or ‘overall”.

L383-384: The intravenous CD3 administration should be better explained. For example: “The intravenous CD3 administration, which is the most commonly used method to define tissue residency and will only bind to circulating cells, …”

L435: Reduction in VSVΔG-GFP infection in BAL cells from pH1N1 infected pigs does not necessarily confirm the inhibitory role of IFI6 specifically. Many other factors could be also involved with creating the antiviral state, they may just not have been detected by you. The only way to confirm the role of IFI6 is to use knockout or suppression/inhibition studies.

Figure 3: The legend in C is too disconnected from it. The figure is very busy. It would be easier to visualize panel D if BAL/PBMC was written on the figure.

Figure 4: Panel C, specify origin of cells used in the figure (HA/NP+Ad-IL-1β pigs).

Reviewer #2: 4. The authors note that all their work was performed on BAL cells that were frozen in an earlier experiment (ref. #44). The authors never provide data on freshly extracted BAL cells as a reference for their studies of the frozen samples. Without this reference the authors should more clearly note that their results are from thawed BAL samples. They should at least note that the process of storage and thawing may have resulted in loss of critical cell subsets.

5. Table S3 would be enhanced with an additional column noting the designated cell subset population, not just the cluster #.

6. For their scRNAseq the authors have used topGO for their Gene Ontology Enrichment (l.556). They note that all results were available in Table S4. However, Table S4 is only, a hard to read listing of GO terms and associations. No % genes identified are noted. This Table should be more clearly presented, using separate columns and adding statistical values that were derived from their data analyses (l.553).

7. The title for Figure 4 implies data interpretation. Add words “test of” in the title - Abundance and test of suppressive function of Tregs. The ordinate labels for the flow histograms in Fig.S4 should note the actual CD antigen as well as the color. That is not noted in the legend either.

8. For Fig 5 the exact cell clusters used to define each cell subset should be noted both in the text and figure legends. The titles for the groups analyzed, e.g. Ad-HA/NP, should be as large as the A B C denoting the graphs. Fig.5 has gene expression on a red background for macrophages; a salmon or purple color would be better. Using the red/blue color for gene names is good for gene regulation; bolding gene names would help to read the names.

9. The authors conclude that addition of Tregs (CD4+CD25high) to the BAL nearly abolished cytokine production by Tregs. Have they now reviewed their clusters [or reanalyzed BAL samples] to identify where that very high CD25+ cell subset actually cluster so that it can be measured?

10. Fig. 5D,E should be a separate figure and enlarged; they are almost impossible to read. The GO terms should be in bold lettering. The figure notes >5 genes are required to be included, but what about % of the GO term genes are differentially expressed for each comparison?

11. Fig 5D notes GO terms not individual genes. On l.341 “… NEBULA identified 10 DE genes shared by cell-types Figure 5D).” What are those 10 genes?

PLOS authors have the option to publish the peer review history of their article (what does this mean?). If published, this will include your full peer review and any attached files.

Reviewer #1: No

Reviewer #2: No
---

## [Decision Letter · Decision Letter 1]

7 Jun 2024

Dear Gerner,

We are pleased to inform you that your manuscript 'Single-cell analysis reveals lasting immunological consequences of influenza infection and respiratory immunisation in the pig lung' has been provisionally accepted for publication in PLOS Pathogens.

Best regards,

Amy L. Hartman, PhD

Academic Editor

PLOS Pathogens

Benhur Lee

Section Editor

PLOS Pathogens

Michael Malim

Editor-in-Chief

PLOS Pathogens

orcid.org/0000-0002-7699-2064

Reviewer Comments (if any, and for reference):

Reviewer's Responses to Questions

**Part I - Summary**

Reviewer #1: My comments were addressed.

Reviewer #2: Much improved manuscript and figures and tables

**Part II – Major Issues: Key Experiments Required for Acceptance**

Reviewer #1: (No Response)

Reviewer #2: good revisions

**Part III – Minor Issues: Editorial and Data Presentation Modifications**

Reviewer #1: (No Response)

Reviewer #2: l.350 pH1H1 should be pH1N1

TS5 would be improved if column W were moved to column B so readers would quickly see the gene name

PLOS authors have the option to publish the peer review history of their article (what does this mean?). If published, this will include your full peer review and any attached files.

Reviewer #1: No

Reviewer #2: No

---

## [Editor Report · Acceptance letter]

26 Jun 2024

Dear Dr Gerner,

We are delighted to inform you that your manuscript, "Single-cell analysis reveals lasting immunological consequences of influenza infection and respiratory immunization in the pig lung," has been formally accepted for publication in PLOS Pathogens.

Best regards,

Michael Malim

Editor-in-Chief

PLOS Pathogens

orcid.org/0000-0002-7699-2064